# Soot Formation in Spherical Diffusion Flames

Sergey M. Frolov [1,*], Vladislav S. Ivanov [1], Fedor S. Frolov [1], Pavel A. Vlasov [1], Richard Axelbaum [2], Phillip H. Irace [3], Grigoriy Yablonsky [4] and Kendyl Waddell [5]

[1] Semenov Federal Research Center for Chemical Physics of the Russian Academy of Sciences, 119991 Moscow, Russia
[2] McKelvey School of Engineering, Washington University in St. Louis, St. Louis, MO 63130-4899, USA
[3] Mechanical Engineering & Materials Science, Washington University in St. Louis, St. Louis, MO 63130-4899, USA
[4] Chemistry Department, Washington University in St. Louis, St. Louis, MO 63130-4899, USA
[5] Department of Fire Protection and Engineering, College Park, University of Maryland, Baltimore, MD 20742, USA
* Correspondence: smfrol@chph.ras.ru

**Abstract:** In the period from 2019 to 2022, the joint American–Russian space experiment (SE) Flame Design (Adamant) was carried out on the International Space Station (ISS). The purpose of the joint SE was to study the mechanisms of control of soot formation in a spherical diffusion flame (SDF) formed around a porous sphere (PS), and the radiative extinction of the SDF under microgravity conditions. The objects of this study were "normal" and "inverse" SDFs of gaseous ethylene in an oxygen atmosphere with nitrogen addition at room temperature and pressures ranging from 0.5 to 2 atm. A normal flame is a flame formed in an oxidizing atmosphere when fuel is supplied through the PS. An inverse flame is a flame formed in a fuel atmosphere when an oxidizer is introduced through the PS. This article presents the results of calculations of soot formation in normal and inverse SDFs. The calculations are based on a one-dimensional non-stationary model of diffusion combustion of gases with detailed kinetics of ethylene oxidation, supplemented by a macrokinetic mechanism of soot formation. The results indicate that soot formation in normal and inverse SDFs is concentrated in the region where the local C/O atomic ratio and local temperature satisfy the conditions $0.32 < C/O < 0.44$ and $T > 1300$–1500 K.

**Keywords:** space experiment; microgravity; spherical diffusion flame; ethylene; numerical simulation; soot formation; radiative flame extinction

**MSC:** 80A21; 80A25; 80A30

## 1. Introduction

Soot formed during combustion of a hydrocarbon fuel is considered, on the one hand, as among the main environmental pollutants, and on the other hand, as an undesirable by-product of combustion, which increases the thermal load on the walls of the combustion chambers of energy-converting devices [1,2]. Therefore, much attention is paid to the study of the mechanisms of soot formation during combustion of hydrocarbons, and the development of principles for arranging combustion without soot formation is considered among the most important tasks of combustion science [3,4]. There are several recent topical review articles in the literature on the issues of soot formation in stationary gaseous diffusion flames obtained in burners with coflow and counterflow of fuel and oxidizer [5–11]. The threshold local temperature for the onset of soot formation in diffusion flames, $T_c$, has been found to be approximately $T_c \approx 1300$ K [12], 1400 K [13], 1450–1600 K [14], 1250 K [15], and 1650 K [16]. In [17], it was proposed to study the features of diffusion combustion of gases using a spherical diffusion flame (SDF) around a porous sphere (PS) under microgravity conditions. Under such conditions, the structure of a diffusion flame can be described by a relatively simple one-dimensional (1D) model and the influence of various

factors (fuel and/or oxidant dilution with inert gases, etc.) on soot formation and other aspects of diffusion combustion can be studied. In this case, either a combustible gas or an oxidizing agent can be supplied to the PS, and the space surrounding the PS can be, respectively, filled with either an oxidizer or a fuel gas. After ignition, in the former case, the so-called normal flame is formed around the PS, and in the latter case, an inverse flame is formed. Studies of SDF in drop towers for a limited time (2.2 s) show [17–21] that such flames are non-stationary: their size increases with time, and the temperature decreases, which leads to radiative extinction. The radiative extinction of the SDF can be sudden or gradual with fluctuations in temperature and other flame parameters [22]. In [23], a simplified soot formation model is used to describe soot formation in the SDF, which contains three irreversible reactions: the fuel oxidation reaction with the formation of a generalized radical, the soot formation reaction during the interaction of fuel with the generalized radical, and the soot oxidation reaction. Calculations according to the model show that soot formation in the SDF can be influenced by adding an inert gas to the fuel. In [24], based on the calculations of 1D normal and inverse SDFs of ethylene, it is concluded that the threshold local temperature at which soot formation in the flame stops is approximately equal to $T_c \approx 1305$ K. In this case, the threshold local C/O atomic ratio in the mixture is approximately equal to $(C/O)_c \approx 0.53$. In other words, soot is formed only where both the local temperature and the local C/O atomic ratio exceed these threshold values. Note that for the onset of soot formation in homogeneous flames, the threshold local value of the C/O atomic ratio, $(C/O)_c$, must exceed 0.6–0.7 [1] or 0.5 [11].

The development of detailed kinetic models of the soot formation process is currently being actively pursued worldwide. Soot formation is a complex multistage process, including the stages of formation of soot particle nuclei from products formed in the gas phase, their surface growth and coagulation, their activation and deactivation, and oxidation. The main remaining problem is the detailed mechanism for the formation of nuclei of future soot particles. In this regard, the recent work [25] must be noted, which analyzes in great detail all currently existing approaches to describing the process of formation of soot particle nuclei with their possible structure and lifetime, as well as thermodynamic and kinetic aspects of the process of formation of soot particle nuclei.

In the pioneering work on detailed kinetic modeling of the soot formation process [26] a HACA mechanism (H-abstraction-acetylene-addition) was proposed aimed at explaining the rapid growth of soot particle surface via the growth of polyaromatic hydrocarbons (PAHs). The primary role in the HACA mechanism is attributed to the activation of the growing hydrocarbon fragment by the detachment of an H atom and active site formation, while an acetylene molecule attaches to this site and introduces two carbon atoms into the fragment. Taking into account the structural restrictions at acetylene molecule addition, this must form a five-membered ring of carbon atoms, which can then be transformed to a six-membered ring. The main consequences of the HACA mechanism have been confirmed by numerous computational and experimental works. Further development of the mechanism led to the understanding that reacting particle activation can occur not only by splitting off an H atom, but also by its addition as well as its migration inside the particle. As for the nucleation of soot particles, the process of condensation of two pyrene molecules (monomers) consisting of four fused aromatic rings was proposed. However, the attractive van der Waals forces for the two pyrene molecules are too weak for the dimer to survive till the next collision. Therefore, the concept of rotational excitation of one of the fragments was proposed to stabilize the resulting complex. At present, the hypothesis of the formation of a bridging chemical bond in the dimer, which stabilizes it, is more popular.

High-resolution atomic force microscopy confirmed the decisive role of polyaromatic structures in the formation of soot particle nuclei [27]. Aromatic structures with one or several aromatic rings as well as structures with five-membered rings, aliphatic side chains, and linear polyene structures were experimentally determined. The formation of PAHs by the HACA reaction sequence is accompanied by the formation of five-membered rings.

Experiments showed the presence of such rings in the PAH precursor of nuclei and young soot particles.

An important question is whether smaller aromatic structures, even those with one or two aromatic rings, are involved in the formation of soot particle nuclei. Recently, a positive answer to this question was given in several works [28,29]. However, there is still no answer to the question of how the formation of a soot particle nucleus starts and how this process proceeds.

A unified detailed reaction mechanism (DRM) of soot formation was proposed in [30]. This DRM is based on a detailed reaction scheme of soot formation and oxidation. It includes the mechanisms of formation of PAHs, polyenes, soot precursors due to condensation of polyaromatic and polyene molecules, soot particle growth according to the HACA mechanism and polyene molecule addition, the mechanism of acetylene pyrolysis and pure carbon cluster formation, as well as the reactions of hydrocarbon (up to n-hexadecane) oxidation. The main advantage of the DRM is that it satisfactorily describes all the available experimental data on the soot yield during pyrolysis and partial oxidation of various hydrocarbons obtained in kinetic shock tubes. The other advantage of the DRM is that it operates with parameters of the condensed phase such as the particle diameter, mean particle diameter, particle number density, and particle size distribution function. As for the size distribution function, it should be emphasized that bimodal distribution functions of soot particles are often presented in the literature. Bimodality also appears in the DRM [30], if one takes into account the nuclei particles, and not just soot particles. The boundary between them in the DRM is still conditional, but in the future, with further improvement of the model and the emergence of new experimental data on the bimodality of the particle size distribution function, it will be possible to refine these boundaries, which will probably clarify the situation of what is considered a nucleus and what is a soot particle.

At long residence times, the process of physical aggregation (adhesion) of primary spherical soot particles occurs simultaneously with the coagulation process. Secondary soot particles with a certain fractal dimension are formed. The determination of the size of such particles is a difficult task.

In the period from 2019 to 2022, the joint American–Russian Space Experiment (SE) Flame Design (Adamant) was conducted on the International Space Station (ISS). The purpose of the joint SE was to study the mechanisms of control of soot formation in the SDF formed around a PS and the radiative extinction of the SDF under microgravity conditions. The objects of the study were normal and inverse SDFs of gaseous ethylene in an oxygen atmosphere with nitrogen addition at room temperature and pressures from 0.5 to 2 atm [31,32]. The purpose of this work is to simulate the evolution of normal and inverse SDFs observed in the SE and determine the conditions for the onset of soot formation.

## 2. Materials and Methods

### 2.1. Statement of the Problem

Figure 1 shows a photograph of the PS with radius $r_s$ = 3.2 mm, porosity $\varphi$, permeability $\kappa$, and characteristic size of solid skeleton $d$, made of a material with density $\rho_s$, heat capacity $c_s$ and thermal conductivity $\lambda_s$, mounted on a gas supply tube with a passage area $S_{in}$ (inner diameter 1.5 mm) (Figure 1a) and the schematic of the computational domain (Figure 1b). The volume of the chamber $V$, in which the assembly is placed, is approximately equal to 90 L (radius of the outer wall $r_\infty$ = 288 mm). At the initial time $t$ = 0, the entire chamber is filled with a nitrogen-diluted oxidizer or nitrogen-diluted combustible gas (hereinafter, external gas) with temperature $T_0$, pressure $P_0$, and species mass fractions $Y_{i0}$ (hereinafter, index $i$ refers to a specific species of the mixture). At $t$ > 0, a combustible gas or oxidizer (hereinafter referred to as the supplied gas) with constant mass fractions of the species $Y_{i,in}$ is supplied through the gas supply tube and PS with a constant mass flow rate $G_{in}$ (hereinafter, index *in* refers to the gas supply section). The supplied gas, passing through PS pores, displaces the initial external gas from it and enters the chamber. All experiments were performed with ethylene, a relatively simple individual hydrocarbon

that burns with soot formation due to the presence of a double carbon bond in its structure. The kinetics of ethylene oxidation and combustion is well studied. The gas supply through the PS is accompanied by molecular mixing of the supplied and external gases and the formation of a combustible mixture in the vicinity of the PS, which is ignited by an external ignition source (hot wire) at time $t_{ign}$. After a certain transition period, an SDF is formed around the PS. In the experiment, the flame radius is determined by the average size of the luminous zone. In calculations, the flame radius is defined as the radial coordinate of the maximum gas temperature $r_f$, with the origin of the coordinate in the PS center. During combustion, the position of the flame may change: the flame radius may both increase and decrease. Generally speaking, depending on the governing parameters of the problem ($G_{in}$, $Y_{i,in}$, $Y_{i0}$, etc.), situations are possible when $r_f > r_s$ and $r_f \leq r_s$. Combustion in a flame is accompanied not only by chemical energy release, molecular and turbulent transport of mass, momentum, and energy, but also by the processes of radiative heat removal to the surrounding space. Radiation is emitted mainly by triatomic molecules ($H_2O$ and $CO_2$) and diatomic molecules ($N_2$ and $O_2$), as well as soot, which can form during combustion.

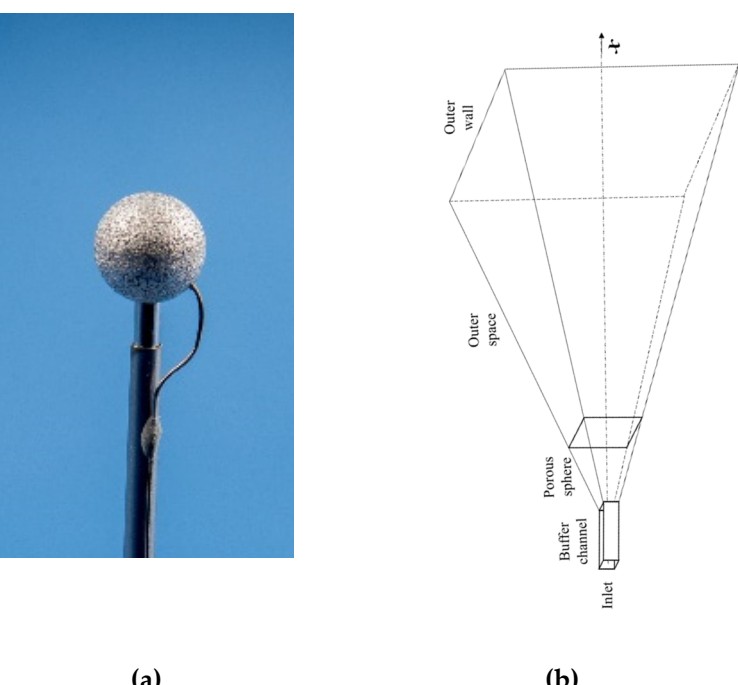

**(a)** **(b)**

**Figure 1.** Photograph of the porous sphere mounted on a gas supply tube (**a**) and the schematic of the computational domain (**b**).

The physical and mathematical model must take into account all most important processes in the described phenomenology. Since the Navier–Stokes equations are valid for both non-porous and porous media, for solving the problem we apply the general three-dimensional (3D) time-dependent Reynolds-Averaged Navier–Stokes (RANS), energy, and species conservation equations for a multicomponent reacting mixture:

$$\rho \frac{\partial U_i}{\partial t} + \rho U_j \frac{\partial U_i}{\partial x_j} = -\frac{\partial p}{\partial x_i} + \frac{\partial}{\partial x_j}\left[\tau_{ij} + \tau_{ij}^t\right]; \tag{1}$$

$$\rho \frac{\partial I}{\partial t} + \rho U_j \frac{\partial I}{\partial x_j} = \rho \dot{Q} + \frac{\partial p}{\partial t} + \frac{\partial}{\partial x_j}\left[\left(\tau_{ij} + \tau_{ij}^t\right)U_j\right] + \frac{\partial}{\partial x_j}\left(q_j + q_j^t\right) + \Omega; \tag{2}$$

$$\rho \frac{\partial Y_l}{\partial t} + \rho U_j \frac{\partial Y_l}{\partial x_j} = \rho \dot{w}_l + \frac{\partial}{\partial x_j}\left[\rho\left(j_{lj} + j_{lj}^t\right)\right], \tag{3}$$

where $x_j$ ($j$ = 1, 2, 3) is the Cartesian coordinate; $\rho$ is the mean gas density; $p$ is the mean pressure; $U_i$ is the $i$th component of the mean velocity vector; $\tau_{ij}$ is the tensor of viscous stresses; $\tau_{ij}^t$ is the tensor of turbulent (eddy) stresses; $I = H + 0.5 \sum_i U_i^2$ is the mean total enthalpy ($H$ is the mean static enthalpy); $q_j$ is the molecular heat flux; $q_j^t$ is the turbulent (eddy) heat flux; $Y_l$ ($l$ = 1, ..., $N$) is the mean mass fraction of the $l$th species ($N$ is the total number of species in the mixture); $\dot{w}_l$ and $\dot{Q}$ are the mean sources of mass and energy due to chemical transformations; $\Omega$ is the heat source/sink other than that of chemical nature; $j_l$ and $j_{lj}^t$ are the molecular and turbulent mass fluxes of the $l$th species, respectively. The turbulent fluxes of momentum, energy, and species in Equations (1)–(3) can be modeled, e.g., within the framework of a certain turbulence model. The governing equations are supplemented with the thermal and caloric equations of state, the relationships for the fluxes and source terms, the reaction mechanism of fuel oxidation and soot formation, as well as by the initial and boundary conditions.

According to the phenomenology, the following simplifying assumptions can be adopted:

1. The gas supply tube does not affect the evolution of the SDF.
2. All processes are spherically symmetric.
3. The porous medium in the flow region can be modeled by flow resistance according to the Darcy law and heat exchange with the fluid according to the Newton law, i.e., the porous medium can be represented by added momentum and heat sources, $\left( \frac{\partial P}{\partial x_i} \right)_s$ and $\Psi_s$, respectively, in the governing equations. In addition, since the porous medium reduces the volume accessible for fluid, the local flow velocity, $U_i$, and superficial velocity inside the porous medium, $u_i$, are coupled by the undirected porosity value $\varphi$: $U_i = \varphi u_i$.
4. The structural and thermophysical parameters of the PS material are constant.
5. PS absorbs thermal radiation of soot, $H_2O$, $CO_2$, $N_2$ and $O_2$; thermal radiation of PS is negligible.
6. Catalytic and gas-phase reactions in the PS are absent.
7. The gas flow is laminar.
8. The gas mixture obeys the ideal-gas thermal and caloric equations of state; gas thermophysical properties are variable.
9. The effect of thermodiffusion is negligible.
10. Soot is an equivalent gas with the molecular mass of atomic carbon, when simulating soot reactions.
11. Soot particles are the clusters of 20–25 carbon atoms, have the corresponding constant size, and do not coagulate.
12. The radiation heat flux is caused solely by soot, $H_2O$, $CO_2$, $N_2$ and $O_2$ emittance.
13. The outer wall of the chamber is impermeable, isothermal, and non-catalytic.

Assumptions 1 and 2 allow one to represent the computational domain in the form of a central sector with a small opening angle divided into control volumes in the radial direction. Assumptions 3 to 6 allow one to write the 1D energy conservation equation for the porous medium in the form:

$$\rho_s c_s \frac{\partial T_s}{\partial t} = \lambda_s \frac{\partial}{\partial x_j} \left( \frac{\partial T_s}{\partial x_j} \right) + \Omega_s; \tag{4}$$

where $T_s$ is the temperature of the PS; $\Omega_s$ is the heat source/sink for the PS. Moreover, assumptions 3 and 5 mean that the term $\Omega_s$ in Equation (4) contains only two contributions: $\Psi_s$ and the radiation absorption $\Omega_{sg}$, whereas assumption 4 implies that the PS radius $r_s$ is constant. Following assumption 7, the turbulent fluxes in Equations (1)–(3) can be omitted, i.e., $\tau_{ij}^t = q_j^t = j_{lj}^t = 0$. Assumption 8 is conventional and implies the validity of the equations of state:

$$p = \rho R T \sum_{l=1}^{N} \frac{Y_l}{W_l} \tag{5}$$

and

$$H_l = H_l^0 + \int_{T^0}^{T} c_{p,l} dT \tag{6}$$

where $W$ is the molecular mass; $R$ is the universal gas constant; $T$ is the temperature; $H_l^0$ is the standard enthalpy of formation of the $l$th species at temperature $T^0$; $c_{p,l}$ is the specific heat at constant pressure. Assumption 9 means that molecular fluxes $q_j$ and $j_{lj}$ can be determined as:

$$q_j = -\left(\lambda \frac{\partial T}{\partial x_j}\right), j_{lj} = -D_l \frac{\partial Y_l}{\partial x_j}, l = 1, \ldots, N-1 \tag{7}$$

where $\lambda$ is the thermal conductivity and $D_l$ is the effective diffusion coefficient of the $l$th species in the mixture [33].

Keeping in mind that existing theoretical knowledge cannot fully explain the occurrence of soot particles, one is faced with a dilemma when simulating soot formation. On the one hand, numerical modeling of the flame seems to require that, in order to match the experimentally observed time instant of the appearance of soot particles, the initiating PAH should be the size of pyrene, and dimerization should be considered as an irreversible process, possibly with a rate reduced by less than an order of magnitude [34]. On the other hand, one cannot fully explain the apparent irreversibility of such a process. In addition, experiments in shock tubes, when the reaction time of the system is limited to one to two milliseconds, convincingly indicate that much smaller molecular fragments with one or two aromatic rings should participate in the formation of soot particle nuclei. Therefore, a simple macrokinetic mechanism [35] of soot formation and oxidation including four irreversible overall reactions is implemented herein:

$$C_2H_2 + C_2H_2 = C + C + C_2H_4 \tag{I}$$

$$C + CO_2 = CO + CO \tag{II}$$

$$C + H_2O = H_2 + CO \tag{III}$$

$$C + OH = HCO \tag{IV}$$

where soot C is considered as a gaseous species in the reaction mechanism with its own mass fraction, $Y_{soot}$, according to assumption 10. This mechanism uses acetylene $C_2H_2$ as a precursor of soot C. Except for soot C, all other substances involved in the reactions of soot formation and oxidation are included in the DRM of ethylene oxidation [36] containing 48 species and 209 reversible elementary reactions. The kinetic parameters of reactions (I)–(IV), the pre-exponential factor, $A_k$, activation energy $E_k$, and the temperature exponent in the expression for the rate of the $k$th reaction, are presented in Table 1.

**Table 1.** Macrokinetic mechanism of soot formation.

| Reaction | $A_k$, [L, mol, s] | $E_k/R$, [K] | $n_k$ |
|:---:|:---:|:---:|:---:|
| I | $2 \times 10^{16}$ | 40,000 | 0 |
| II | $1 \times 10^{15}$ | 40,000 | 0 |
| III | $1 \times 10^{15}$ | 40,000 | 0 |
| IV | $1 \times 10^{12}$ | 0 | 0 |

Note that these parameters were determined using the thoroughly tested DRM of soot formation [30]. In the DRM, soot particle nuclei are formed in processes involving a radical or two polyaromatic radicals and a stable polyaromatic molecule, although the contribution of purely radical reactions is very small. In the case of pyrolysis of acetylene and ethylene, when the PAH concentration is relatively low compared to the pyrolysis and oxidation of other hydrocarbons, the nuclei of soot particles are formed from polyene-like fragments. Thus, depending on the specific hydrocarbon, either the polyaromatic or the polyene pathways for the formation of soot particle nuclei dominate in the DRM.

Parameters $A_k$ and $E_k$ in Table 1 were determined from the condition of the best agreement between the results of calculations for the soot yield obtained on the basis of DRM [30] and on the basis of the macrokinetic mechanism. For estimating the effect of soot radiation it is conditionally assumed (assumption 11) that soot particles possess the specific (per unit mass) emitting surface, $S_{soot} = 6/(d_{soot}\rho_{soot})$ (here $d_{soot}$ is the conditional soot particle size, $\rho_{soot}$ is the soot density), which is directly connected to the soot mass fraction $Y_{soot}$. Thus, the simplest radiation model is used without radiation reabsorption and scattering by soot particles.

Assumption 12 implies that the source term $\Omega$ in Equation (2) contains the contribution $\Omega_{soot}$ proportional to $S_{soot}$. Assumption 12 also implies that the term $\Omega$ in Equation (2) contains the contribution $\Omega_g$ proportional to the local instantaneous volume fractions of $H_2O$, $CO_2$, $N_2$ and $O_2$. Assumption 13 is conventional.

The 3D governing Equations (1)–(4) are now greatly simplified and can be solved in 1D approximation using an available CFD code. The explicit form of the source terms entering Equations (1)–(4) are as follows:

$$\dot{w}_l = W_l \sum_{k=1}^{L} \left( v''_{l,k} - v'_{l,k} \right) A_k T^{n_k} exp \left( -\frac{E_k}{RT} \right) \prod_{j=1}^{N} \left( \frac{Y_j \rho}{W_j} \right)^{v'_{j,k}} \tag{8}$$

$$\dot{Q} = \sum_{l=1}^{N} H_l \dot{w}_l \tag{9}$$

$$\Omega = \Omega_{soot} + \Omega_g + \Psi_s \tag{10}$$

$$\Omega_{soot} = \sigma S_{soot} Y_{soot} \rho \left( T^4 - T_0^4 \right) \tag{11}$$

$$\Omega_g = \sigma p \sum_{l=1}^{4} a_l(T) X_l \left( T^4 - T_0^4 \right) \tag{12}$$

$$\Psi_s = \alpha_s S_{PS} \left( T - T_s \right) \tag{13}$$

$$\Omega_s = \Psi_s + \Omega_{sg} \tag{14}$$

$$\Omega_{sg} = \delta_s \varepsilon_s \int \left( \Omega_{soot} + \Omega_g \right) dV \tag{15}$$

$$\left( \frac{\partial p}{\partial x_i} \right)_s = \left( \frac{\partial p}{\partial x} \right)_s = \frac{\mu}{\kappa} u \tag{16}$$

where $L$ is the total number of chemical reactions in the gas; $v''_{l,k}$ and $v'_{l,k}$ are the stoichiometric coefficients of the $l$th species in the mixture, which is a reactant and product in the $k$th reaction, respectively; $\sigma$ is the Stefan–Boltzmann constant; $a_l$ and $X_l$ are the emissivity and volume fraction of the $l$th emitting gas; $\alpha_s$ is the heat transfer coefficient between gas and PS; $S_{PS} = 6(1 - \varphi)/d$ is the specific surface area of the PS; $\varepsilon_s$ is the coefficient of radiation absorption by the PS material; $\delta_s$ is the delta function used to account for radiation absorption only on the PS surface; $\mu$ is the dynamic viscosity of the gas. Equation (16) is used according to recommendations [37]. The values of the coefficients $a_l$ for $H_2O$ and $CO_2$ are taken from the polynomials in [38], for $N_2$ and $O_2$ $a_l$ is independent of the gas temperature and is assumed to be equal to 0.1. The dynamic viscosity $\mu$ and thermal conductivity $\lambda$ of the gas, as well as the effective diffusion coefficients of species in the gas mixture, $D_l$, and specific heats $c_{p,l}$ are calculated by the formulae presented in [39].

Initial conditions:

$$t = 0, \ 0 \le x \le r_\infty : \ p = p_0, \ T = T_0, \ Y_l = Y_{l0} \ (l = O_2, N_2), \ Y_l = 0 \ (l \ne O_2, N_2) \tag{17}$$

Boundary conditions:

$$t > 0,$$

$$x = 0 : \rho U S_{in} = G_{in}, \ T = T_0, \ Y_l = Y_{l0} \ (l = C_2H_4, N_2), \ Y_l = 0 \ (l \ne C_2H_4, N_2) \tag{18}$$

$$x = r_\infty : U = 0, \ T = T_0, \ \frac{\partial Y_l}{\partial x} = 0, l = 1, \ldots, N \qquad (19)$$

The task of calculations is to determine the spatial structure of the SDF and its evolution in time, calculate the time dependence of such characteristics of the SDF as the flame radius $r_f$, the PS temperature $T_s$, the maximum gas temperature $T_f = T_{g,max}$ (flame temperature), as well as the total soot mass:

$$m_{soot,\Sigma}(t) = \int_{r_s}^{r_\infty} 2\pi \rho_{soot} Y_{soot}(t) x^2 dx$$

cumulative rate of soot formation:

$$\dot{m}_{soot,\Sigma}(t) = \frac{dm_{soot,\Sigma}(t)}{dt},$$

and integral soot mass fraction $Y_{soot,\Sigma}$:

$$Y_{soot,\Sigma}(t) = \frac{1}{V} \int_{r_s}^{r_\infty} 2\pi Y_{soot}(t) x^2 dx$$

It is worth noting that, in accordance with the theory, an important parameter of diffusion flames is the stoichiometric mixture fraction $0 < Z_{st} < 1$ determined by the mass fractions of the oxidizer $Y_O$ and fuel $Y_F$ in the burner:

$$Z_{st} = \frac{Y_O}{Y_O + \eta Y_F}$$

where $\eta = \nu_O W_O / \nu_F W_F$, and $\nu_O = 3$ and $\nu_F = 1$ are the stoichiometric coefficients in the overall reaction equation $C_2H_4 + 3O_2 = 2CO_2 + 2H_2O$, whereas $W_O = 32$ kg/kmol and $W_F = 28$ kg/kmol are the molecular masses of oxygen and ethylene. Large values of $Z_{st}$ correspond to flames with oxygen excess, and small values correspond to flames with fuel excess. Normal flames in an atmosphere of oxygen diluted with nitrogen, with the supply of undiluted ethylene to the PS, correspond to small values of $Z_{st}$. Inverse flames in an atmosphere of ethylene diluted with nitrogen, with the supply of undiluted oxygen to the PS, correspond to large values of $Z_{st}$. It is generally believed that flames with low $Z_{st}$ values are more prone to soot formation than flames with large $Z_{st}$ values [40,41].

*2.2. Numerical Solution*

The computational domain of Figure 1b contains a planar buffer channel with a cross-sectional area $S_{in}$, PS, and an outer space bounded by a wall. The planar buffer channel of length $r_0 = 10$ mm is used to provide undisturbed boundary conditions. The buffer channel is immersed in the PS so that the area of the inlet section in the PS is equal to $S_{in}$. Symmetry conditions are satisfied on the side surfaces of the computational domain. The spherical part of the computational domain in Figure 1b is represented by a spherical segment with a solid angle of 3°. The entire computational domain is divided into 1333 control volumes compressed towards the outer surface of the PS, which ensures the grid independence of the obtained results.

The set of governing Equations (1)–(4) with additional relations (5)–(16) is solved by the control volume method using the segregated SIMPLE/PISO algorithm [32]. Convective transfer in the law of conservation of mass is approximated by the central difference and in the law of conservation of momentum by the total variation diminishing (TVD) scheme with the MINMOD limiter. For the finite-volume approximation of all other equations, the standard UPWIND scheme of the first order is used.

Before calculating the flame dynamics, an ignition procedure was developed. The ignition procedure required the weakest possible effect on the flow field caused by the ignition induced pressure rise. Preliminary calculations showed that the stronger the ignition, the later the arising flow field became insensitive to ignition The resulting ignition

procedure is as follows. In a time interval of 0.2–0.3 s after the start of gas supply to the PS, in the region $x \in [r_s + 0.0005 \text{ m}, r_s + 0.0060 \text{ m}]$, the mixture temperature is instantaneously changed to $T_{ign}$ = 1300 K. These time intervals and regions allow the formation of a small portion of flammable premixed fuel–oxidizer composition. An ignition temperature of 1300 K is approximately the minimal temperature required for mixture ignition.

Calculations are made for the following values of governing parameters: $T_0$ = 293 K, $p_0$ = 0.1 MPa (unless otherwise specified), $S_{in}$ = 5·$10^{-10}$ m$^2$, $r_s$ = 0.0032 m, $r_\infty$ = 0.288 m, $\varphi$ = 0.5, $\kappa$ = $10^{-13}$ m$^2$; $d$ = $10^{-5}$ m; $\varepsilon_s$ = 0.8, $\rho_s$ = 4000 kg/m$^3$, $c_s$ = 650 J/(kg·K), $\lambda_s$ = 5 W/(m·K), $d_{soot}$ = 2 nm, $\rho_{soot}$ = 2000 kg/m$^3$, and $N$ = 49, $L$ = 213.

## 3. Results of Experiments and Calculations

Tables 2 and 3 show the conditions for some selected experiments with normal and inverse flames, respectively. The experiments were carried out both with pure ethylene and with ethylene diluted with nitrogen. Ethylene and nitrogen were supplied either to the PS (normal flames) or to the combustion chamber (inverse flames) from different cylinders without premixing. The instantaneous flow rates of both gases were controlled by flow meters. This approach made it possible to arbitrarily control the degree of ethylene dilution. During the SE, no samples of soot or gas were taken. All information on the experiment was sent to the Mission Control Center via communication channels.

**Table 2.** Conditions of some normal flames.

| Flame | Combustion Chamber | | Porous Sphere | | | $p$, atm |
|---|---|---|---|---|---|---|
| | $X_{O_2}$ | $X_{N_2}$ | $X_{C_2H_4}$ | $X_{N_2}$ | $G_{in}$, mg/s | |
| 19115B1 | 0.203 | 0.797 | 1.000 | 0.000 | 0.660 | 1.020 |
| 19206L6 | 0.194 | 0.806 | 0.288 | 0.712 | 0.660 | 1.010 |
| 19171D4 | 0.363 | 0.637 | 1.000 | 0.000 | 1.372 | 1.010 |
| 19189K1 | 0.363 | 0.637 | 0.476 | 0.524 | 1.372 | 1.310 |
| F10 | 0.380 | 0.620 | 1.000 | 0.000 | 1.224 | 1.239 |
| F02 | 0.391 | 0.609 | 0.288 | 0.712 | 1.800 | 1.190 |
| F08 | 0.386 | 0.614 | 0.288 | 0.712 | 3.603 | 1.263 |
| F05 | 0.400 | 0.600 | 0.288 | 0.712 | 4.514 | 1.250 |
| 19156C2 | 0.366 | 0.634 | 1.000 | 0.000 | 2.529 | 1.040 |
| 19142J3 | 0.356 | 0.644 | 1.000 | 0.000 | 2.529 | 0.990 |
| 19150N1 | 0.296 | 0.704 | 0.168 | 0.832 | 4.885 | 1.010 |
| 19150G3 | 0.338 | 0.662 | 0.288 | 0.712 | 8.779 | 1.050 |
| 19175A3 | 0.391 | 0.609 | 1.000 | 0.000 | 1.960 | 1.270 |
| 19206A5 | 0.207 | 0.793 | 0.288 | 0.712 | 8.779 | 1.010 |
| 19206G1 | 0.205 | 0.795 | 1.000 | 0.000 | 2.529 | 1.010 |
| 19206G4 | 0.201 | 0.799 | 1.000 | 0.000 | 0.822 | 1.010 |
| 19206L4 | 0.195 | 0.805 | 0.288 | 0.712 | 2.835 | 1.010 |
| 19115M4 | 0.193 | 0.807 | 0.476 | 0.524 | 1.380 | 1.020 |
| 19123F1 | 0.206 | 0.794 | 0.490 | 0.510 | 2.640 | 1.010 |
| 19123F2 | 0.206 | 0.794 | 0.489 | 0.511 | 2.640 | 1.010 |
| 19123F3 | 0.205 | 0.795 | 0.490 | 0.510 | 2.640 | 1.010 |
| 19123L1 | 0.202 | 0.798 | 1.000 | 0.000 | 2.510 | 1.010 |
| 19123L2 | 0.201 | 0.799 | 1.000 | 0.000 | 2.510 | 1.010 |
| 19150N1 | 0.351 | 0.649 | 0.168 | 0.832 | 4.820 | 1.040 |
| 19189J3 | 0.378 | 0.622 | 0.502 | 0.498 | 5.010 | 1.300 |
| 19200H3 | 0.285 | 0.715 | 0.131 | 0.869 | 4.430 | 1.020 |
| 19115F1 | 0.204 | 0.796 | 0.292 | 0.708 | 2.180 | 1.040 |
| 19123A2 | 0.209 | 0.791 | 1.000 | 0.000 | 1.620 | 1.000 |
| 19123A3 | 0.208 | 0.792 | 1.000 | 0.000 | 1.620 | 1.000 |
| 19123A4 | 0.208 | 0.792 | 1.000 | 0.000 | 1.620 | 1.000 |
| 19123C1 | 0.207 | 0.793 | 0.290 | 0.710 | 4.460 | 1.000 |

**Table 3.** Conditions for some inverse flames.

| Flame | Combustion Chamber | | Porous Sphere | | | $p$, atm |
| | $X_{C_2H_4}$ | $X_{N_2}$ | $X_{O_2}$ | $X_{N_2}$ | $G_{in}$, mg/s | |
|---|---|---|---|---|---|---|
| 21328D1 | 0.257 | 0.743 | 0.212 | 0.788 | 10.05 | 1.03 |
| 21349M3 | 0.270 | 0.730 | 0.212 | 0.788 | 9.11 | 1 |
| 22018H2 | 0.097 | 0.903 | 0.497 | 0.503 | 6.37 | 1.01 |
| 22018J1 | 0.096 | 0.904 | 0.318 | 0.682 | 9.73 | 1.01 |
| 22018G3 | 0.098 | 0.902 | 0.850 | 0.150 | 7.89 | 1.01 |
| 22018G2 | 0.099 | 0.901 | 0.850 | 0.150 | 5.90 | 1.01 |
| 22018G1 | 0.099 | 0.901 | 0.850 | 0.150 | 3.90 | 1 |
| 21328N5 | 0.080 | 0.920 | 0.850 | 0.150 | 2.27 | 0.96 |
| 22035J2 | 0.096 | 0.904 | 0.850 | 0.150 | 5.90 | 0.51 |
| 21340M1 | 0.121 | 0.879 | 0.850 | 0.150 | 9.22 | 1.01 |
| 21340M2 | 0.274 | 0.726 | 0.262 | 0.738 | 8.8 | 1.01 |
| 21349N3 | 0.251 | 0.749 | 0.212 | 0.788 | 9.10 | 0.52 |
| 21349N4 | 0.246 | 0.754 | 0.212 | 0.788 | 10.03 | 0.52 |
| 22018B1 | 0.168 | 0.832 | 0.850 | 0.150 | 4.7 | 1 |
| 22024F1 | 0.187 | 0.813 | 0.412 | 0.588 | 9.16 | 1.01 |
| 22024B1 | 0.189 | 0.811 | 0.850 | 0.150 | 5.9 | 1.01 |

Figure 2 shows examples of images of a sooty normal flame (Figure 2a) and a sooty inverse flame (Figure 2b) [42,43]. Normal and inverse sooty flames differ in that in a normal flame, soot forms inside the flame, and in an inverse flame, it forms outside the flame. The reason for this difference is the different arrangement of spatial zones with a composition enriched with fuel: in a normal flame, fuel (ethylene) enters through the PS, which is heated by the heat flux from the flame, thermally decomposes and gives rise to soot formation nuclei inside the flame, while in the inverse flame, fuel (ethylene) is located outside the flame and, being heated by the flame-induced heat flux, thermally decomposes in the outer vicinity of the flame, giving rise to soot formation. The latter is clearly seen in Figure 2b, where a yellow glow is observed outside of the flame.

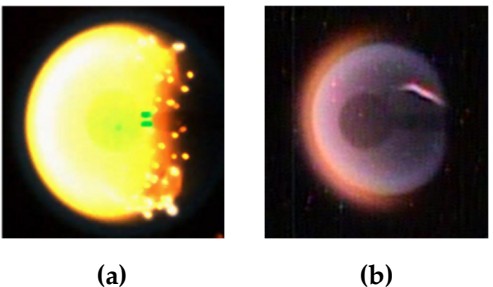

(**a**)                                    (**b**)

**Figure 2.** Photographs of the spherical diffusion flames of ethylene in the Flame Design (Adamant) SE under microgravity conditions on the ISS: (**a**) sooty normal flame; (**b**) sooty inverse flame [42,43].

Figures 3 and 4 show examples of the measured and calculated time histories of the flame radius and temperature for normal and inverse flames, which undergo radiative extinction ~48 and ~30 s after ignition, respectively. The symbols correspond to the experiment, and the curves correspond to the calculation according to the model described above. Figure 3 shows that both the normal and inverse flames slowly move away from the PS with time. The measured and calculated flame temperatures agree satisfactorily with each other (see Figure 4), and the radiative extinction of flames occurs when the temperature of both the normal and inverse flames decrease with time to a value of approximately 1200 K. It is important to note that the model satisfactorily describes both the dynamics of changes in the flame size and the dynamics of changes in flame temperature. Similar findings were reported in [44].

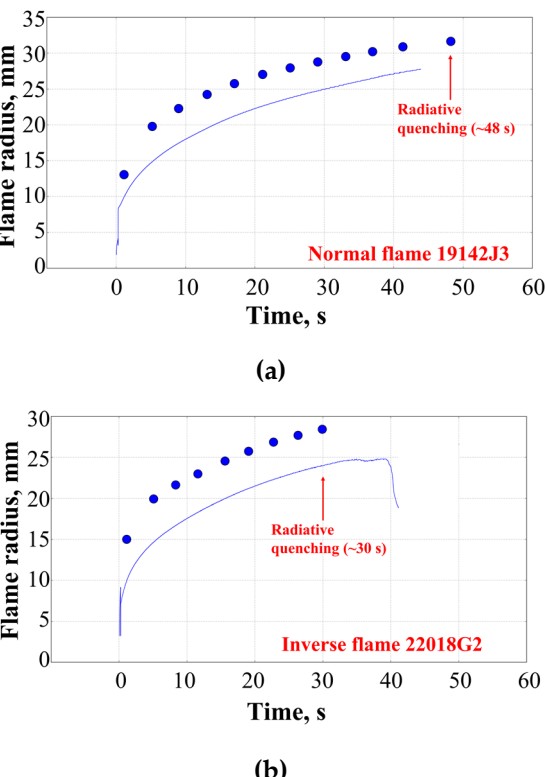

**Figure 3.** Examples of the time histories of the flame radius: (**a**) normal flame; (**b**) inverse flame. Symbols correspond to the experiment, and the curves correspond to the calculation.

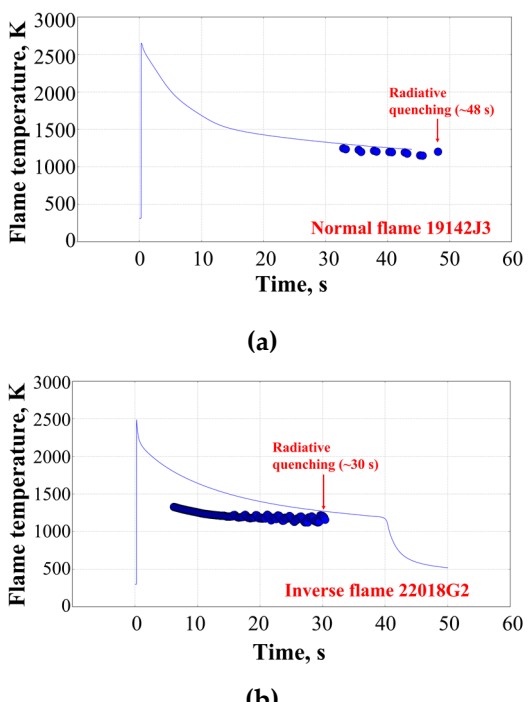

**Figure 4.** Examples of the time histories of the flame temperature: (**a**) normal flame; (**b**) inverse flame. Symbols correspond to the experiment, and the curves correspond to the calculation.

Among the most important new results of the SE is the registration of quasi-stationary inverse diffusion flames. Figure 5 compares the measured and calculated time histories of the size and temperature of such a quasi-stationary inverse flame (the flame temperature

was not measured in this experiment). It can be seen that, with time, the position of the inverse flame stabilizes at a certain distance from the PS, and its temperature, after a certain period of decline, is maintained almost constant at a level of 1600–1700 K.

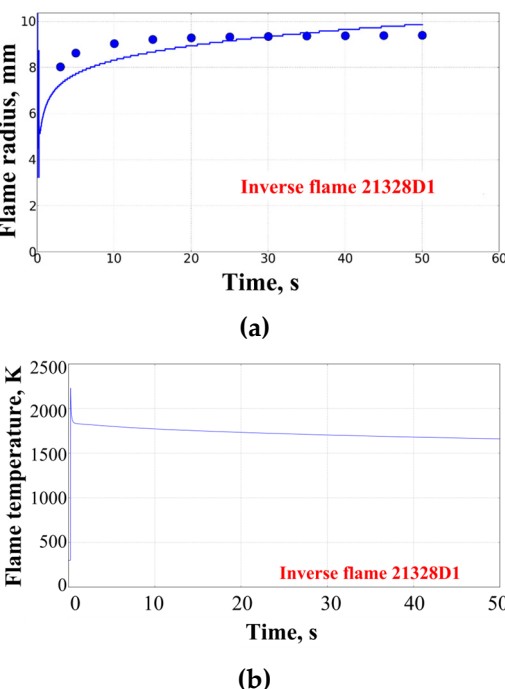

(a)

(b)

**Figure 5.** Examples of time histories of the size (**a**) and temperature (**b**) of a quasi-stationary inverse flame. Symbols correspond to the experiment, and the curves correspond to the calculation.

Figure 6 collects all the calculated time histories of the temperature of normal and inverse flames from Tables 2 and 3, which have undergone radiative extinction. The radiative extinction of the flame is caused by an increase in its surface due to expansion and a corresponding increase in heat losses due to radiation, leading to a decrease in the flame temperature. Note that the radiative extinction of all considered flames was registered both in experiments and in calculations. Regardless of the type of flame (normal or inverse), the composition of the gas supplied to the PS, and the composition of the atmosphere around the PS, the flame quenches when its temperature drops to ~1200 K. At a lower temperature, the reaction rates become insufficient to sustain combustion, and the flame quenches. This result is of great fundamental importance for the theory of diffusion combustion.

Figure 7 shows typical calculated structures of normal (Figure 7a) and inverse (Figure 7b) flames 20 s after ignition. The temperature is normalized by the maximum value. The soot mass fractions are increased by a large scale factor ($10^4$ in Figure 7a and $10^6$ in Figure 7b) in order to consider the details of the soot spatial distribution. It can be seen that at the point at which the temperature reaches its maximum value, the concentrations of ethylene and oxygen are almost zero, and the concentrations of intermediate and final reaction products reach values close to their maxima. In both flames, at some distance from the flame, the mass fractions of all intermediate and final reaction products drop to zero, whereas the mass fractions of the initial substances and the gas temperature are restored to their initial values. Despite the similarity of the structures of the normal and inverse flames, they fundamentally differ in the spatial distributions of soot. If in a normal flame, soot is present only inside the flame (between the PS and the temperature maximum), then in an inverse flame, soot is present only outside the flame (outside the temperature maximum).

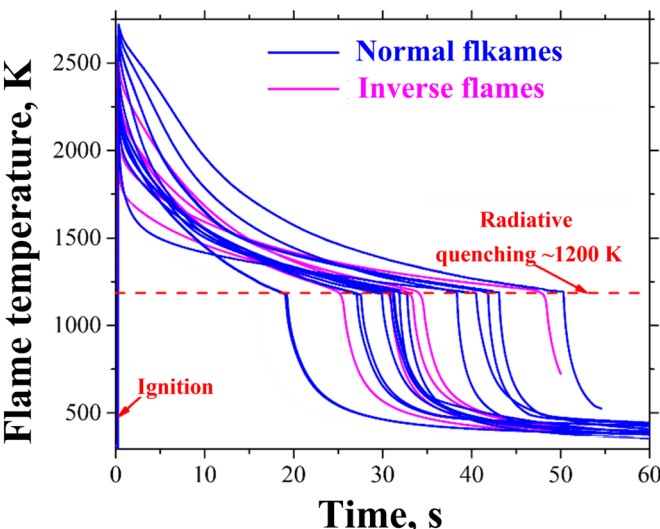

**Figure 6.** Calculated time histories of the temperature of normal and inverse flames undergoing radiative extinction.

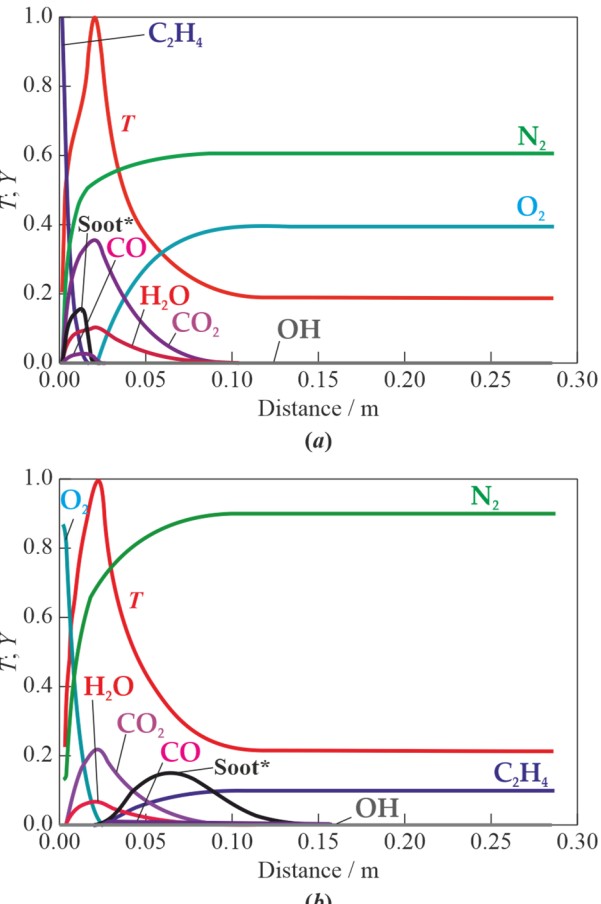

**Figure 7.** Typical calculated structures of normal (**a**) and inverse (**b**) diffusion flames in 20 s after ignition. Soot mass fractions are increased by a factor of $10^4$ (**a**) and $10^6$ (**b**).

It is interesting to consider the temporal dynamics of soot yield in normal and inverse flames from Tables 2 and 3. Figure 8 shows the dynamics of the calculated total soot mass $m_{soot,\Sigma}$ versus time for flames with different values of the $Z_{st}$. Each flame corresponds to a fixed value of $Z_{st}$. Normal flames are represented by blue dots, while inverse flames are represented by red dots. Each figure corresponds to a specific moment in time from 1 to

50 s after ignition. The figure corresponding to the time of 1 s is repeated at the end of the list for the convenience of comparing the soot yields at the times of 50 s and 1 s. From the last two figures (see the dashed blue and red circles) it can be seen that the total mass of soot formed at the initial moments of time (during ignition) in all normal flames (small $Z_{st}$ values) gradually decreases (blue dots move down vertically), and in inverse flames either almost does not change (large values of $Z_{st}$), or increases (small values of $Z_{st}$). Thus, the dynamics of soot formation obtained with the simplified model as a whole does not agree with the expected trend: normal flames with small $Z_{st}$ values turn out to be less prone to soot formation than inverse flames with large $Z_{st}$ values, although inverse flames with small $Z_{st}$ values show the expected tendency to a greater propensity to soot formation.

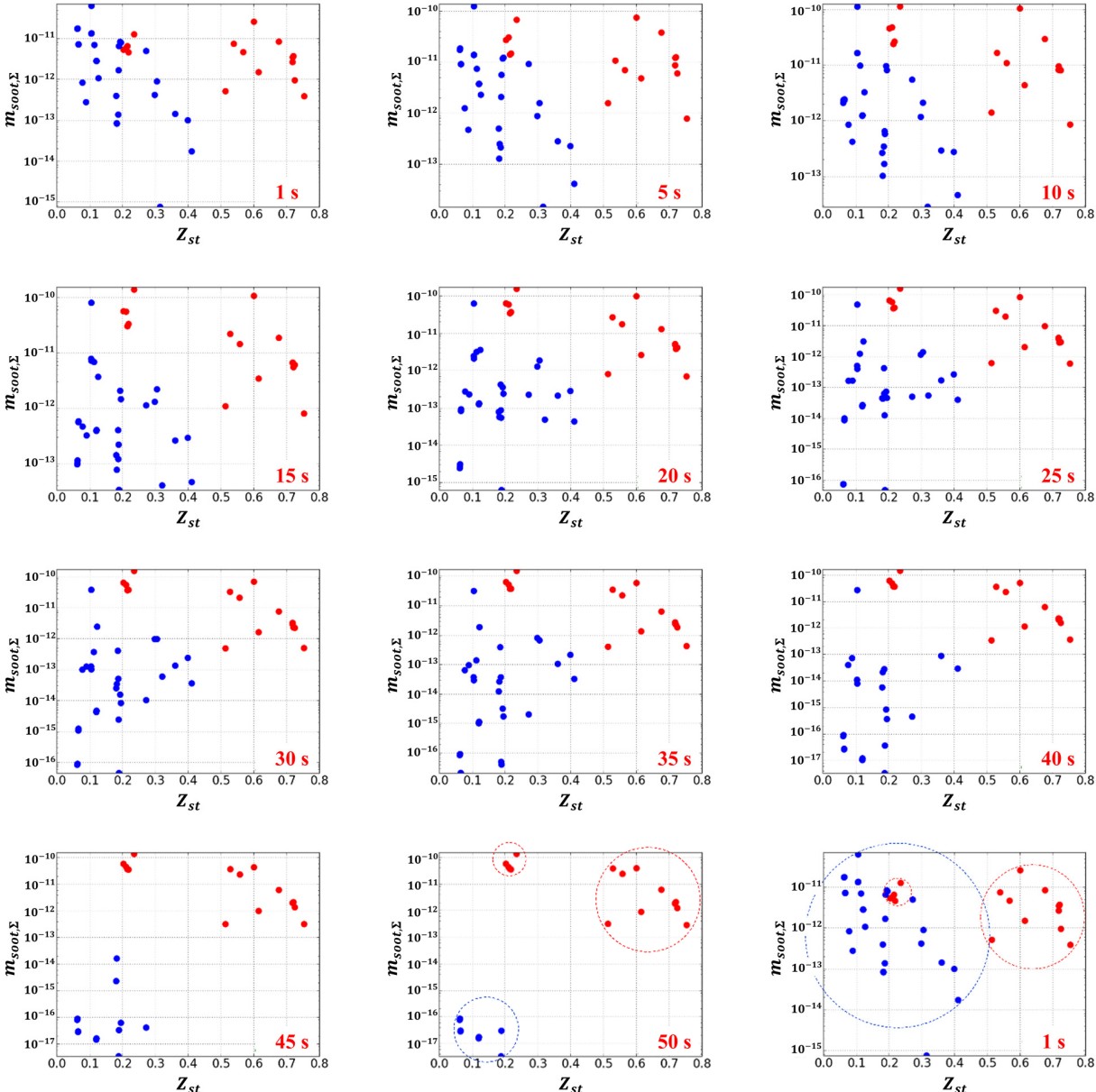

**Figure 8.** Dynamics of change in the calculated total soot mass $m_{soot,\Sigma}$ versus time for flames with different values of $Z_{st}$. Normal flames are represented by blue dots, while inverse flames are represented by red dots.

The results obtained mean that soot formation under terrestrial conditions and under microgravity conditions is different. The latter is related to the flow structure in a diffusion

flame under microgravity conditions. In expanding normal flames, soot particles formed inside the flame due to the pyrolysis of ethylene are carried by a radial flow towards the flame to the high temperature region and are oxidized by intermediate and final reaction products. In this case, due to the gradual decrease in the flame temperature, the rate of soot formation in the vicinity of the flame decreases. Apparently, in expanding normal flames with small $Z_{st}$ values, the soot formed during ignition almost completely disappears due to oxidation. In inverse expanding flames, the soot particles formed outside the flame as a result of ethylene pyrolysis are carried by a radial flow away from the flame to the low temperature region, where the rates of soot oxidation reactions are much lower than in the flame. In this case, as in a normal flame, due to the gradual decrease in the flame temperature, the rate of soot formation in the vicinity of the flame decreases. Apparently, in inverse flames with large $Z_{st}$ values the soot formed during ignition is virtually not oxidized according to the simplified model. As for inverse flames with small $Z_{st}$ values, these flames have a quasi-stationary structure, relatively small dimensions, and high flame temperatures (see Figure 5). In the vicinity of such flames, soot continuously forms and the formed soot is carried by the flow to the low temperature region and accumulates.

In addition to Figure 8, Figure 9 shows the results of calculations of the normalized cumulative rate of soot formation $\dot{m}_{soot,\Sigma}(t)$ in normal and inverse flames listed in Tables 2 and 3 as a function of the instantaneous local gas temperature at the point of maximum soot mass fraction in the flame structure (see Figure 7). The cumulative rate of soot formation in Figure 9 is normalized by the absolute maximum of the cumulative rate of soot formation $\dot{m}_{soot,\Sigma}(max)$ for each flame. As before, blue and red dots correspond to normal and inverse flames, respectively. Figure 9 makes it possible to determine the temperature in the flame structure at which the cumulative rate of soot formation changes sign (from plus to minus). A positive value of $\dot{m}_{soot,\Sigma}(t)$ means that soot formation dominates in the flame structure at a given time. A negative value of $\dot{m}_{soot,\Sigma}(t)$ means that soot oxidation dominates in the flame structure at a given time. The temperature at which the sign of the quantity $\dot{m}_{soot,\Sigma}(t)$ changes from plus to minus can be interpreted as the threshold temperature at which soot formation stops. It follows from Figure 9 that at the beginning of the process (immediately after ignition), soot formation dominates (the points are in the upper half-plane). Indeed, due to the fact that during ignition there are regions with a partially mixed composition enriched with fuel and a high temperature, soot formation can be very significant. This means that ignition strongly affects the dynamics of soot formation in these flames. Therefore, to better understand the specific features of soot formation during spherical diffusion combustion, one must consider the process time at which the role of soot formed during ignition becomes insignificant. The flame temperature decreases with time, and the normalized cumulative rates of soot formation also decrease tending to zero in the temperature range 1300–1500 K. With a further decrease in temperature, soot oxidation is seen to dominate (the points move to the lower half-plane). Thus, for normal flames, the threshold temperature at which soot formation stops lies in the range of 1400–1500 K. For inverse flames, this threshold value is somewhat lower: in the range of 1300–1400 K. It should be emphasized that Figure 9 does not reflect the tendency of certain flames to form soot, since the value $\dot{m}_{soot,\Sigma}(t)$ is normalized by the maximum cumulative rate of soot formation $\dot{m}_{soot,\Sigma}(max)$. Here, only the temperature at which this quantity changes sign is important. Based on the data in Figure 9, soot formation in inverse flames stops at lower temperatures than in normal flames, i.e., inverse flames appear to be generally more prone to soot formation than normal flames. The latter is consistent with the conclusions made from considering Figure 8.

To confirm the established patterns of soot formation in normal and inverse SDFs, we have plotted Figure 10, where at the time moments 2, 5 and 10 s after ignition, the red dots show the positive values of the cumulative rate of soot formation, $\dot{m}_{soot,\Sigma}(t) > 0$, and the blue dots show the negative values, $\dot{m}_{soot,\Sigma}(t) < 0$, depending on the values of the

atomic ratio C/O and the temperature at the point of the maximum soot concentration in the flame structure. On the one hand, it can be seen that red dots appear at a minimum temperature of 1260 K after 2 s (Figure 10a), 1300 K after 5 s (Figure 10b), and 1310 K after 10 s (Figure 10c) after ignition. Blue dots appear at a maximum temperature of 1470 K in 2, 5 and 10 s after ignition. If one assumes that the "memory depth" of the ignition stage is approximately 10 s, then for the entire family of normal and inverse flames, the cumulative rate of soot formation becomes positive when the temperature exceeds 1300–1500 K (blue vertical band in Figure 10c). On the other hand, it can be seen that red dots appear at a minimum value of C/O = 0.31 in 2 s (Figure 10a), C/O = 0.32 in 5 s (Figure 10b) and C/O = 0.33 in 10 s (Figure 10c) after ignition. The maximum C/O values at which red dots still exist are C/O = 0.58 in 2 s (Figure 10a), C/O = 0.49 in 5 s (Figure 10b), and C/O = 0.44 in 10 s (Figure 10c) after ignition. If one assumes that the "memory depth" of the ignition stage is ~10 s, then for the entire family of normal and inverse flames, the cumulative rate of soot formation becomes positive when the C/O atomic ratio lies in the range from 0.32 to 0.44 (red horizontal bar). Interestingly, the lower value $(C/O)_c \approx 0.32$ is close to the stoichiometric value $(C/O)_{st} \approx 0.33$ for the ethylene–oxygen mixture, at which the temperature in the flame is maximum.

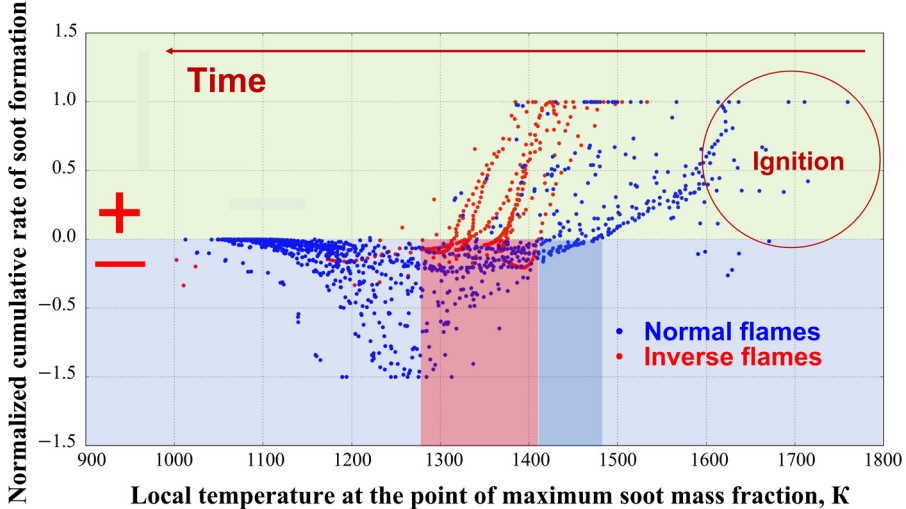

**Figure 9.** The normalized cumulative rate of soot formation $\dot{m}_{soot,\Sigma}(t)$ in normal and inverse flames listed in Tables 2 and 3 as a function of the instantaneous local gas temperature at the point of maximum soot mass fraction in the flame structure.

In [24], based on 1D calculations of the normal and inverse SDFs of ethylene, it was concluded that the threshold local temperature at which soot formation in the flame stops is approximately equal to $T_c \approx 1305$ K, whereas the local C/O atomic ratio in the mixture is approximately equal to $(C/O)_c \approx 0.53$. In other words, soot is formed only where both the temperature and the C/O atomic ratio exceed these values. Figure 11 shows the key graphs from [24]. The maxima of the temperature curves are close to the stoichiometric atomic ratio $(C/O)_{st} = 0.33$, although the maxima of inverse flames are slightly shifted towards lower C/O values. It follows from Figure 11a that the curves for all flames converge at local values of temperature and C/O atomic ratio approximately equal to 1305 K and 0.53, respectively. Figure 11b additionally shows the calculated dependences of the corresponding temperature values ($T_{0.53}$) at C/O = 0.53 and the ratio $(C/O)_{1305}$ at a temperature of 1305 K, showing that for all flames considered in [24] the threshold values of these parameters are approximately the same. Note that the calculations in [24] were performed for the conditions of terrestrial experiments in a falling platform, which provided a relatively short time for observing SDFs (up to 2.2 s only).

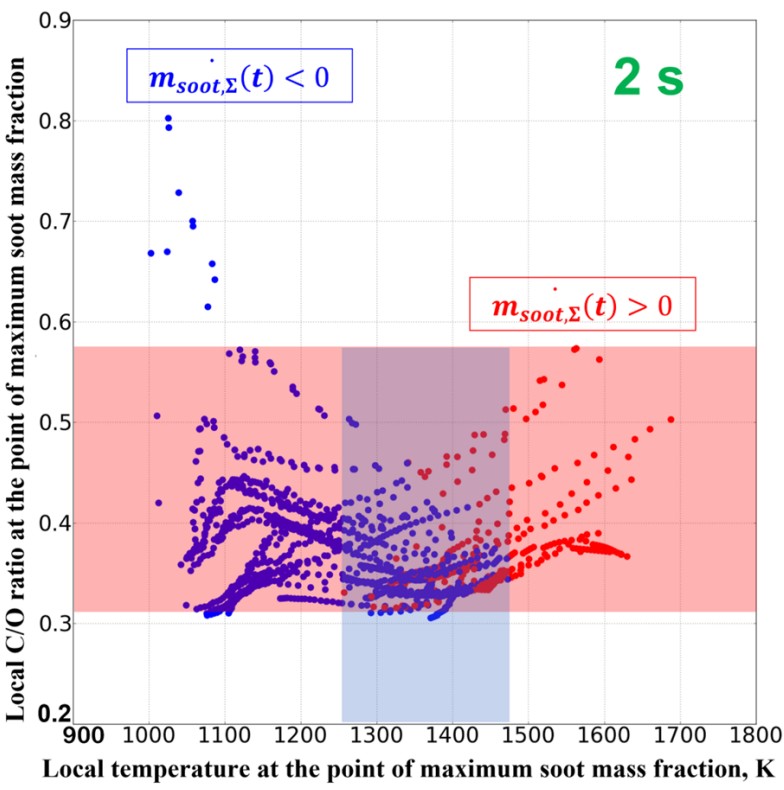

(a)

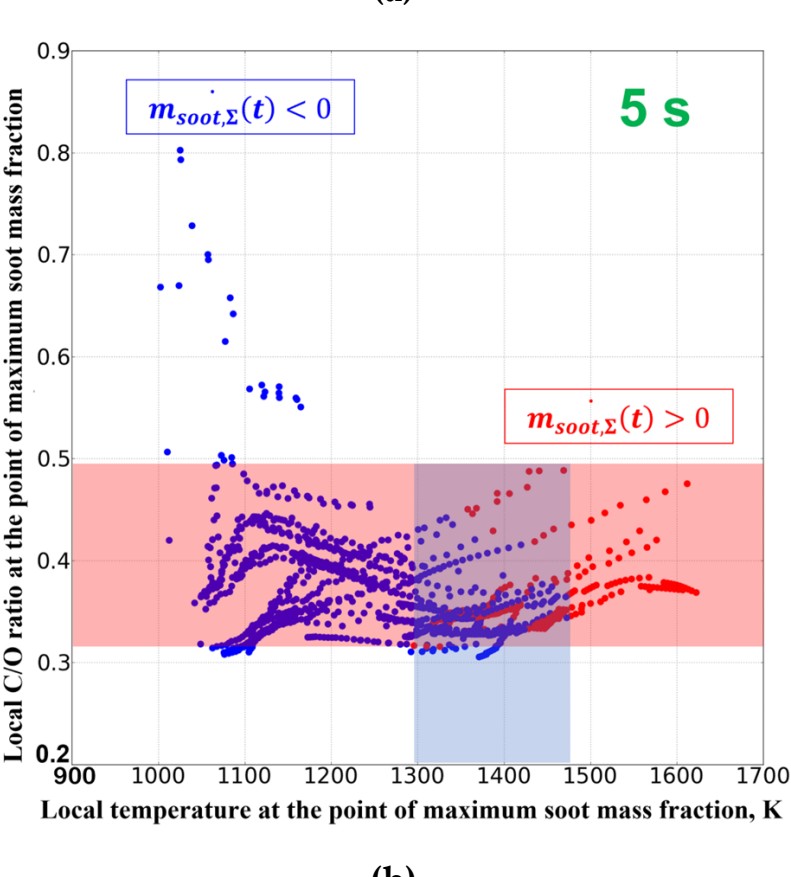

(b)

**Figure 10.** *Cont.*

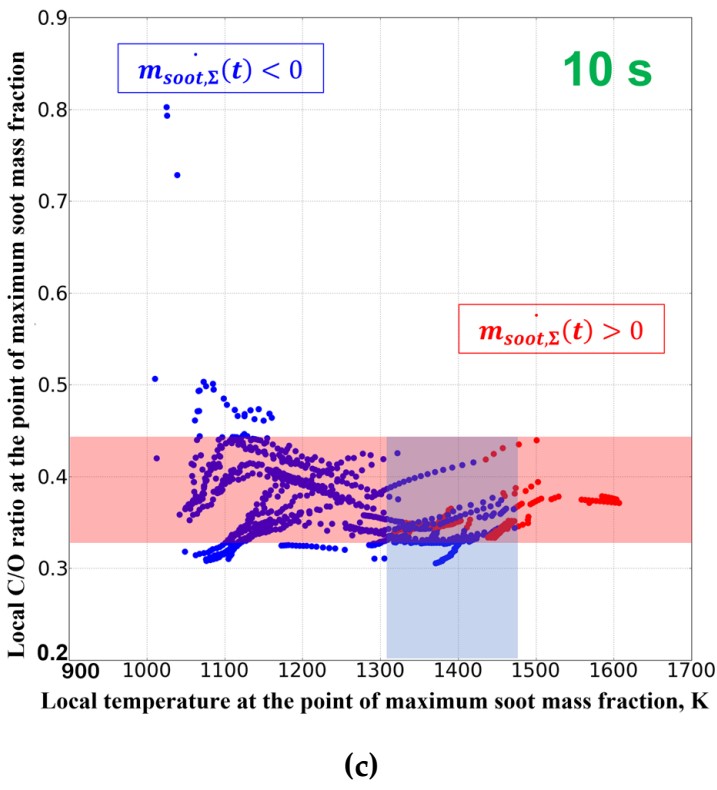

**(c)**

**Figure 10.** Calculated dependences of the cumulative rate of soot formation $\dot{m}_{soot,\Sigma}(t)$ on the C/O atomic ratio and temperature at the point of maximum soot concentration in the flame structure at different times after ignition: (**a**) 2 s; (**b**) 5 s; (**c**) 10 s; red dots: $\dot{m}_{soot,\Sigma}(t) > 0$, and blue dots: $\dot{m}_{soot,\Sigma}(t) < 0$.

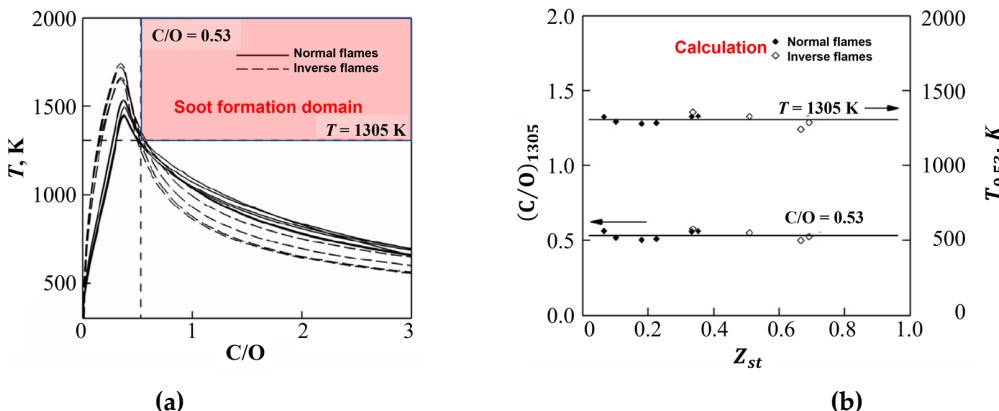

**(a)**          **(b)**

**Figure 11.** Calculated dependences of (**a**) temperature on the local atomic ratio C/O and (**b**) temperature $T_{0.53}$ and atomic ratio $(C/O)_{1305}$ on the stoichiometric mixture fraction $Z_{st}$ in normal and inverse flames in 2 s after ignition [24].

Figure 12 shows the results of our calculations in the form of the same dependences of the gas temperature on the local C/O atomic ratio for normal and inverse flames from Tables 2 and 3 in 2, 10, 20 and 30 s after ignition. Normal and inverse flames are represented by blue and red curves, respectively. As in Figure 11a, the maxima of the temperature curves are close to the stoichiometric atomic ratio $(C/O)_{st} = 0.33$, while the maxima of the inverse flames are slightly shifted towards lower C/O values. With the passage of time, when the maximum temperature decreases to 1200 K, some of the considered flames undergo radiative extinction, and the corresponding curves flatten out. As for the threshold values $T_c \approx 1305$ K and $(C/O)_c \approx 0.53$ obtained in the calculations [24], they turn out to

be inapplicable for SDFs in the Flame Design (Adamant) SE: there is no convergence of the temperature curves at these values in 2 s after ignition (see Figure 12a), although at $T = T_c \approx 1300$ K, the values of the C/O atomic ratio vary from 0.32 to 1.0, i.e., include the value $(C/O)_c \approx 0.53$. These differences can be explained by the closeness of the conditions in [24] to the moment of ignition: the soot formed during the ignition of a partially premixed combustible mixture can affect the result. Indeed, as early as in 10 s after ignition (see Figure 12b), all temperature curves approach each other significantly. The threshold values of the C/O atomic ratio ($0.32 < (C/O)_c < 0.44$) and temperature ($1300 < T_c < 1500$ K) established above (see Figure 9) are shown in Figure 12b,c, and Figure 12d as dashed vertical and horizontal lines, respectively, and the parametric regions $[(C/O)_c, T_c]$ bounded by these curves are colored with red fill. It can be seen that at times of 10, 20, and 30 s after ignition, these parametric regions are realized in the structure of normal and inverse flames with a relatively high temperature, while in the structure of flames with a relatively low temperature, there is no soot formation at these times.

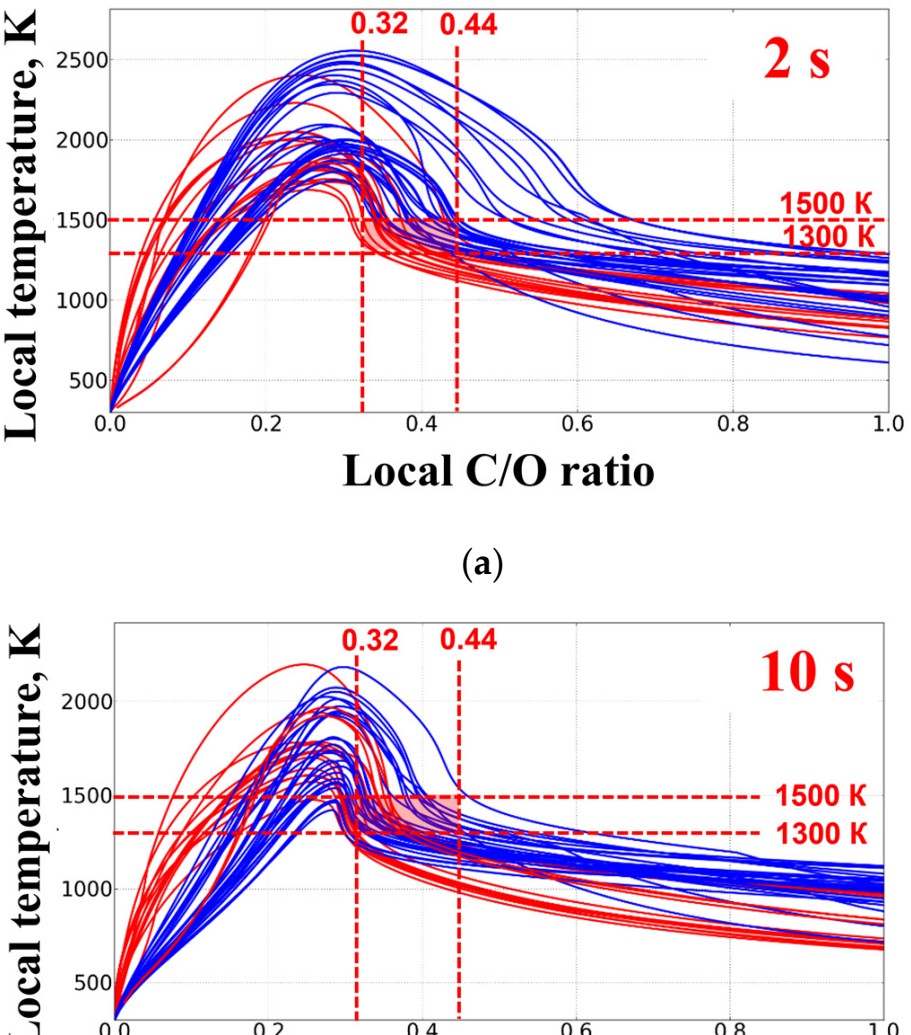

(a)

(b)

**Figure 12.** *Cont.*

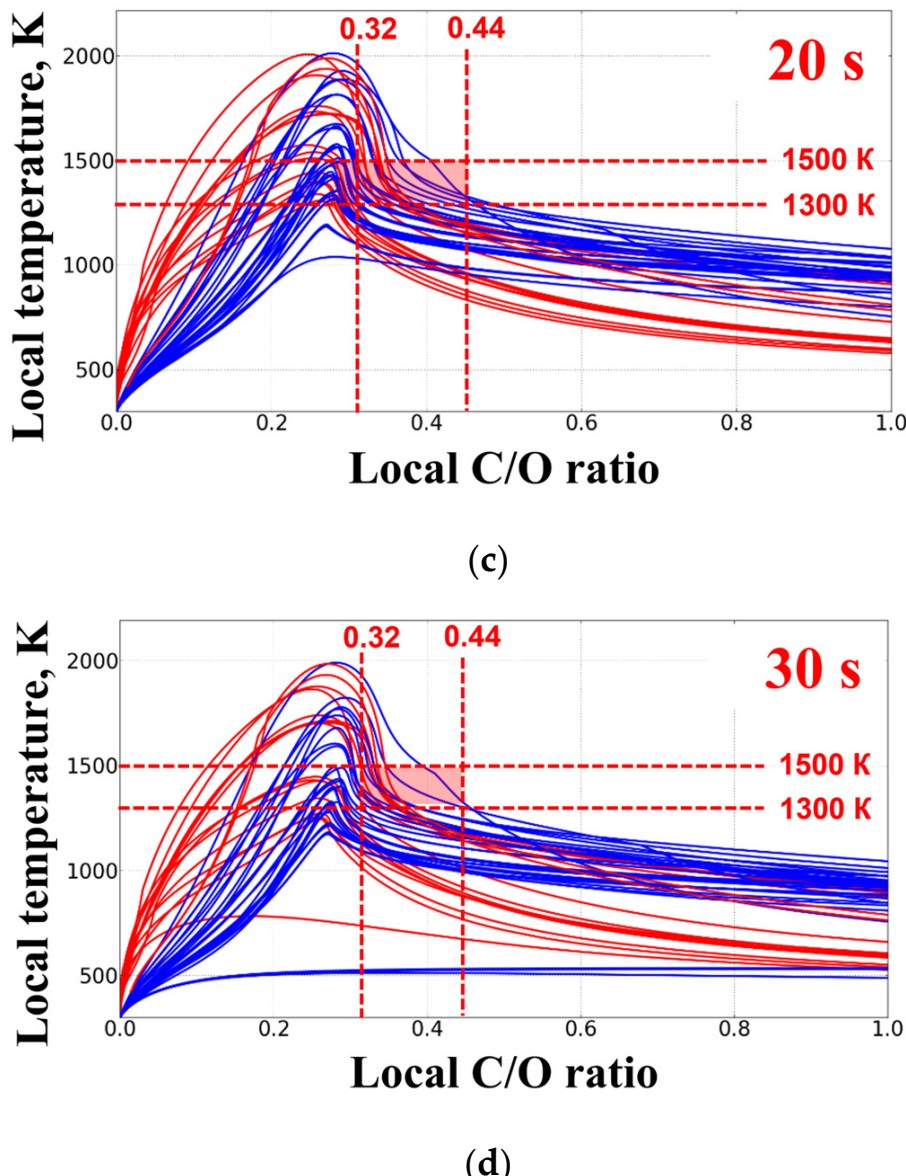

**Figure 12.** Calculated dependences of temperature on the local C/O atomic ratio in normal (blue curves) and inverse (red curves) flames in 2 (**a**), 10 (**b**), 20 (**c**), and 30 s (**d**) after ignition. The regions colored with red fill correspond to the parametric domains of soot formation.

Thus, SDFs with a higher temperature should be more prone to soot formation than flames with a lower temperature. The results in Figure 12 can be interpreted as follows: *soot formation in normal and inverse flames is concentrated in the region where T > 1300–1500 K and 0.32 < C/O < 0.44.* Such a condition for the onset of soot formation in terms of temperature generally agrees with the literature data for gaseous diffusion flames obtained in burners with coflow or counterflow of fuel and oxidizer under the conditions of terrestrial gravity [41,45,46]. As for the conditions of soot formation in terms of the local C/O atomic ratio, the obtained threshold values of C/O turn out to be lower than the known literature data (0.5 < C/O < 0.7) for diffusion flames under the conditions of terrestrial gravity. However, there are data in the literature that soot nanoparticles can also form at a local atomic ratio C/O = 0.4 in homogeneous ethylene–air flames at atmospheric pressure [47].

## 4. Discussion

The data in Figure 12 allow judgement about the tendency of certain flames to form soot. When considering normal flames, the data in Figure 12 can be interpreted as follows.

Near the PS, the C/O atomic ratio is very high; moving away from the PS corresponds to movement along the C/O axis from right to left with a gradual increase in temperature. Soot formation in the flame starts when C/O values become lower than $(C/O)_c \approx 0.44$, and the temperature becomes higher than $T_c = 1300$–$1500$ K. Soot formation ends when C/O reaches a near-stoichiometric value $(C/O)_{st} \approx 0.32$, and the temperature reaches its maximum value. With further movement to the left along the C/O axis, soot formation is impossible due to the high temperature and the presence of free oxygen, which prevents the formation of acetylene, a soot precursor in the macrokinetic mechanism. When considering inverse flames, the data in Figure 12 can be interpreted as follows. Near the PS, the C/O atomic ratio is very low; moving away from the PS corresponds to movement along the C/O axis from left to right with a gradual increase in temperature. Soot formation in the flame starts when C/O exceeds the near-stoichiometric value $(C/O)_{st} \approx 0.32$, and the temperature passes through the maximum: the gas composition in the region beyond the temperature maximum is enriched with fuel, which promotes the formation of acetylene, a soot precursor. Soot formation ends when the value of C/O becomes higher than $(C/O)_c \approx 0.44$, and the temperature becomes lower than $T_c = 1300$–$1500$ K. Such an interpretation of the conditions of soot formation in SDFs makes it possible to propose approaches for reducing the soot yield. Among the simplest approaches is a small partial premixing of fuel and oxidizer despite premixing in terrestrial flames can promote pyrolysis and soot formation [48]. Instead of pure ethylene, a partially premixed mixture of ethylene with oxygen with an atomic ratio, for example, C/O = 5 (normal flame), can be fed into the PS, or an atmosphere with C/O = 5 can be created in the combustion chamber instead of an ethylene atmosphere with C/O $\rightarrow \infty$ (inverse flame). In both cases, in the zone with $(C/O) > (C/O)_{st}$, the formation of acetylene (soot precursor) will be difficult due to the presence of free oxygen. Apparently, such an interpretation of the conditions of soot formation in SDFs is applicable to flames of different fuels, with the only difference being that the lower limit of the C/O interval is determined by the type of combustible gas (for ethylene and propane, this is $(C/O)_{st} \approx 0.33$–$0.30$, and for methane $(C/O)_{st} = 0.25$).

Note that in the macrokinetic mechanism of the formation of soot particles, which is used in this work, some important stages of the soot formation process are absent. The processes of propargyl recombination [40], surface growth of soot particles, and soot particle coagulation are not considered. Therefore, such a model cannot adequately predict the size of soot particles, their size distribution function, and the concentration of soot particles. The model considers very small particles with a diameter of 2 nm, which is several times smaller than the experimental one. Typical sizes of soot particles in various flames are 15–30 nm. Therefore, the macrokinetic mechanism considered in this paper better describes the earliest stages of the soot formation process and worse describes its later stages [49–51]. Although acetylene molecules are not directly precursors of soot particle nuclei, the role of acetylene in the process of soot formation is important, especially in the process of particle surface growth.

One should also take into account that the process of oxidation of fuel-rich mixtures of ethylene leads to the appearance of high concentrations of acetylene molecules along with the formation of molecules of polyaromatic compounds. Therefore, it can be assumed that the nuclei of soot particles are formed in this case along both polyaromatic and polyene pathways. Since the macrokinetic mechanism considers very small soot particles, the role of the process of their oxidation can most likely be overestimated.

All these circumstances must be taken into account when using the macrokinetic mechanism of soot formation and when interpreting the results of calculations. Nevertheless, due to the fact that the microkinetic mechanism is based on the validated DRM of soot formation the predicted general trends could be considered as reliable.

## 5. Conclusions

In this work, the evolution of 31 normal and 16 inverse spherical diffusion flames of ethylene under microgravity conditions was calculated. With the help of calculations, the

conditions of soot formation in such flames were determined. The calculations were based on a one-dimensional non-stationary model of diffusion combustion of gases with the detailed kinetics of ethylene oxidation, supplemented by a macrokinetic mechanism of soot formation. It was shown that soot formation in normal and inverse flames is concentrated in the region where the local temperature and local C/O atomic ratio satisfy the conditions $T > 1300$–$1500$ K and $0.32 < C/O < 0.44$. The condition for the onset of soot formation in terms of temperature generally agrees with the literature data for gaseous diffusion flames obtained in burners with coflow and counterflow of fuel and oxidizer under terrestrial gravity. As for the conditions of soot formation in terms of the values of the local C/O atomic ratio, the obtained threshold values of C/O turn out to be lower than the known literature data (C/O > 0.53) for diffusion flames. The future work will be focused on the further development of the computational approach by introducing a more elaborate model of soot formation including the evolution of the soot particle size distribution function, as well as a more elaborate model of soot radiation including radiation reabsorption and scattering effects.

**Author Contributions:** Conceptualization, S.M.F. and R.A.; methodology, S.M.F. and R.A.; investigation, V.S.I., F.S.F., P.A.V., P.H.I., G.Y. and K.W.; data curation, V.S.I., F.S.F., P.H.I. and K.W.; writing—original draft preparation, S.M.F.; writing—review and editing, S.M.F. and R.A. All authors have read and agreed to the published version of the manuscript.

**Funding:** This research was funded by the Russian Space Agency Roskosmos (Adamant project) and NASA (grant numbers 80NSSC20M0072 and 80NSSC20M0073).

**Data Availability Statement:** Research data can be provided upon request.

**Conflicts of Interest:** The authors declare no conflict of interest.

## Nomenclature

| | |
|---|---|
| $a_l$ | Emissivity of the $l$th emitting gas |
| $A_k$ | Pre-exponential factor |
| $c_{p,l}$ | Specific heat at constant pressure |
| $c_s$ | Solid skeleton heat capacity |
| $(C/O)_c$ | Threshold local C/O atomic ratio |
| $d$ | Characteristic size of solid skeleton |
| $d_{soot}$ | Conditional soot particle size |
| $D_l$ | Effective diffusion coefficient of the $l$th species |
| $E_k$ | Activation energy |
| $\left(\dfrac{\partial p}{\partial x_i}\right)_s$ | Added momentum source in porous medium |
| $G_{in}$ | Inlet mass flow rate |
| $H$ | Mean gas static enthalpy |
| $H_l^0$ | Standard enthalpy of formation of the $l$th species |
| $I$ | Mean gas total enthalpy |
| $j_l$ | Molecular mass flux of the $l$th species |
| $j_{lj}^t$ | Turbulent mass flux of the $l$th species |
| $L$ | Total number of chemical reactions in the gas |
| $m_{soot,\Sigma}$ | Total soot mass |
| $\dot{m}_{soot,\Sigma}$ | Cumulative rate of soot formation |
| $n_k$ | Temperature exponent |
| $N$ | Number of gas species |
| $P$ | Mean gas pressure |
| $P_0$ | Initial pressure |
| $q_j$ | Molecular heat flux |
| $q_j^t$ | Turbulent heat flux |
| $Q$ | Mean source of energy due to chemical transformations |

| | |
|---|---|
| $r_0$ | Length of the buffer channel |
| $r_f$ | Flame radius |
| $r_s$ | Radius of porous sphere |
| $r_\infty$ | Radius of the outer wall of the chamber |
| $R$ | Universal gas constant |
| $S_{in}$ | Passage area of gas supply tube |
| $S_{PS}$ | Specific surface area of the porous sphere |
| $S_{soot}$ | Specific emitting surface area |
| $t$ | Time |
| $t_{ign}$ | Time of ignition |
| $T$ | Temperature |
| $T_0$ | Initial temperature |
| $T^0$ | Standard temperature |
| $T_c$ | Threshold local temperature of soot formation |
| $T_{ign}$ | Ignition temperature |
| $T_s$ | Temperature of porous sphere |
| $u_i$ | Superficial velocity inside the porous medium |
| $U_i$ | The $i$th component of the mean gas velocity vector |
| $V$ | Chamber volume |
| $\dot{w}_l$ | Mean source of mass due to chemical transformations |
| $W$ | Molecular mass |
| $W_O$ | Molecular mass of oxidizer |
| $W_F$ | Molecular mass of fuel |
| $x_j$ | Cartesian coordinate |
| $X_l$ | Volume fraction of the $l$th emitting gas |
| $Y_{i0}$ | Initial species mass fractions |
| $Y_{i,in}$ | Inlet species mass fractions |
| $Y_l$ | Mean mass fraction of the $l$th species |
| $Y_{soot}$ | Soot mass fraction |
| $Y_{soot,\Sigma}$ | Integral soot mass fraction |
| $Z_{st}$ | Stoichiometric mixture fraction |
| $\alpha_s$ | Heat transfer coefficient between gas and porous sphere |
| $\delta_s$ | Delta function |
| $\varepsilon_s$ | Coefficient of radiation absorption by the porous sphere material |
| $\kappa$ | Permeability |
| $\lambda$ | Thermal conductivity of the $l$th species |
| $\lambda_s$ | Solid skeleton thermal conductivity |
| $\mu$ | Dynamic viscosity of gas |
| $\nu_O$ | Stoichiometric coefficient of oxidizer in the overall reaction equation |
| $\nu_F$ | Stoichiometric coefficient of fuel in the overall reaction equation |
| $v'_{l,k}$ | Stoichiometric coefficients of the $l$th species in the reactants of the $k$th reaction |
| $v''_{l,k}$ | Stoichiometric coefficients of the $l$th species in the products of the $k$th reaction |
| $\rho$ | Mean gas density |
| $\rho_s$ | Solid skeleton density |
| $\rho_{soot}$ | Soot density |
| $\sigma$ | Stefan–Boltzmann constant |
| $\tau_{ij}$ | Tensor of viscous stresses |
| $\tau_{ij}^t$ | Tensor of turbulent stresses |
| $\varphi$ | Porosity |
| $\Psi_s$ | Added heat source in porous medium |
| $\Omega$ | Heat source/sink other than that of chemical nature |
| $\Omega_s$ | Heat source/sink for porous sphere |
| $\Omega_{sg}$ | Radiation absorption |

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
