# Peer review of "Soot Formation in Spherical Diffusion Flames"

_mathematics, doi:10.3390/math11020261_

Round 1

Reviewer 1 Report

The authors studied the soot formation in spherical diffusion flame, here are my comments/questions:

1. Is Figure 1 original from this work or adapted from other work?

2. What radiation model did the authors use in this work? Did the authors consider the reabsorption effects and scatting effects which are important for soot radiation?

3. What are the difference of soot formation between micro-gravity environment and normal environment?

4. The font sizes in Figures 7, 9, 11 are too small. The authors may want to modify it.

5. The conclusion part seems too simple. The authors may want to have more discussions in the conclusion.

Author Response

We are grateful to the reviewer for valuable comments. We made our best to follow all the comments. All changes in the revised manuscript are marked in yellow.

  1. Is Figure 1 original from this work or adapted from other work?

Figure 1 is plotted purposely for this manuscript.

  1. What radiation model did the authors use in this work? Did the authors consider the reabsorption effects and scatting effects which are important for soot radiation?

At this stage of the study, we have used the simplest radiation model without radiation reabsorption and scattering by soot particles, see Eq. (11). Actually, the model operates with conditional soot particles, which are needed only for estimating the specific surface area for soot radiation. To address this comment, we have added the following sentence to the manuscript:

“Thus, the simplest radiation model is used without radiation reabsorption and scattering by soot particles.”

Also, in the Discussion section, we have added a reference to our recent article in Atmosphere [45], which studies the evolution of the soot particle size distribution function.

  1. What are the difference of soot formation between micro-gravity environment and normal environment?

The simple macrokinetic model of soot formation of Table 1 was developed by fitting its predictions for the soot yield with the predictions provided by the Detailed Reaction Mechanism (DRM) [26] in the wide range of governing parameters. It is noted in the Introduction, that the DRM satisfactorily describes all the available experimental data on the soot yield during pyrolysis and partial oxidation of various hydrocarbons obtained in kinetic shock tubes. As the DRM is composed of elementary homogeneous and heterogeneous reactions (which are independent of gravity), it must be applicable to both normal and micro-gravity conditions. In view of it, one can assume that our microkinetic model could be equally applicable to these conditions as well.

  1. The font sizes in Figures 7, 9, 11 are too small. The authors may want to modify it.

We have replotted Figures 7, 9, and 11 with larger fonts and changed a little figure captions.

  1. The conclusion part seems too simple. The authors may want to have more discussions in the conclusion.

To address this comment, we have added one paragraph to the Conclusios:

"The future work will be focused on the further development of the computational approach by introducing a more elaborate model of soot formation including the evolution of the soot-particle size distribution function, as well as a more elaborate model of soot radiation including radiation reabsorption and scattering effects."

Reviewer 2 Report

Summary: The paper is clearly written, well organized, and easy to follow. It studied the normal and inverse spherical diffusion flame in order to control soot formation from gaseous ethylene in an oxygen atmosphere with nitrogen addition at room temperature and pressures ranging from 0.5 to 2 atm. 

Specific comments:

The same abbreviation SDF is applied to two terms: spherical diffusion flame and size distribution function. This is confusing to readers. Either change  size distribution function (SDF)  to particle size distribution function (PSDF), or do not abbreviate size distribution function.

Table 3. all the decimal point “.” is mistyped as “,” in columns 3-7 .

Line 353: TVD (total variation diminishing) to total variation diminishing (TVD)

Author Response

We are grateful to the reviewer for valuable comments. We made our best to follow all the comments. All changes in the revised manuscript are marked in green.

Summary: The paper is clearly written, well organized, and easy to follow. It studied the normal and inverse spherical diffusion flame in order to control soot formation from gaseous ethylene in an oxygen atmosphere with nitrogen addition at room temperature and pressures ranging from 0.5 to 2 atm. 

Specific comments:

The same abbreviation SDF is applied to two terms: spherical diffusion flame and size distribution function. This is confusing to readers. Either change  size distribution function (SDF)  to particle size distribution function (PSDF), or do not abbreviate size distribution function.

We have removed the abbreviation for sized distribution function.

Table 3. all the decimal point “.” is mistyped as “,” in columns 3-7 .

We have replaced commas by dots.

Line 353: TVD (total variation diminishing) to total variation diminishing (TVD)

Done.

Reviewer 3 Report

1. Line 29-35: In introduction part, it may be better supplement relevant references in these paragraphs.

2. Line 37: Excessive references are cited here at once, a brief description of the work of these literature can be presented.

3. Line 148, 150: What are the component of combustible gas and oxidizer?

4. Line 160: How the combustible gas is ignited and where the external ignition source located? it may be better to add a brief explanation for this.

5. Line 162: It would be better to add a description for the creation of the coordinate system.

6. The authors give a very detailed calculation of the model, which contains a large number of symbols, and some of which appear frequently in the paper, adding a nomenclature for some key words (such as Zst) might make it easier to read.

Author Response

We are grateful to the reviewer for valuable comments. We made our best to follow all the comments. All changes in the revised manuscript are marked in blue.

  1. Line 29-35: In introduction part, it may be better supplement relevant references in these paragraphs.
  2. Line 37: Excessive references are cited here at once, a brief description of the work of these literature can be presented.

To address comments 1 and 2, we have somewhat rearranged references [1-7] in lines 29-35.

  1. Line 148, 150: What are the component of combustible gas and oxidizer?

The combustible gas and oxidizer can be diluted by nitrogen. To address this comment, we have replaced “oxidizer or combustible gas” by “nitrogen-diluted oxidizer or nitrogen-diluted combustible gas” at the beginning of Section 2.1.

  1. Line 160: How the combustible gas is ignited and where the “external ignition source” located? it may be better to add a brief explanation for this.

The ignition procedure in the calculations is described in detail at the end of Section 2.2. Nevertheless, to address this comment, we have replaced “an external ignition source” by “an external ignition source (hot wire)” in the description of the phenomenon at the beginning of Section 2.1.

  1. Line 162: It would be better to add a description for the creation of the coordinate system.

To address this comment, we have replaced “coordinate of the maximum gas temperature ” by “radial coordinate of the maximum gas temperature , with the origin of the coordinate in the PS center.”

  1. The authors give a very detailed calculation of the model, which contains a large number of symbols, and some of which appear frequently in the paper, adding a nomenclature for some key words (such as Zst) might make it easier to read.

To address this comment, we have added the Nomenclature to the manuscript.

Round 2

Reviewer 1 Report

The paper can be accepted now.